# DebUnc: Improving Large Language Model Agent Communication Via Uncertainty Metrics

## Abstract

To enhance Large Language Model (LLM) capabilities, multi-agent debates have been introduced, where multiple LLMs discuss solutions to a problem over several rounds of debate. However, LLMs often produce incorrect responses that appear confident, which can mislead other agents. This is partly because agents do not express their confidence levels during standard debates. To address this, we introduce DebUnc, a multi-agent debate framework that uses uncertainty metrics to assess agent confidence levels. We adapted the LLM attention mechanism to adjust token weights based on confidence levels and also explored using textual prompts to convey confidence. Our evaluations across various benchmarks show that attention-based methods are particularly effective, and that as uncertainty metrics improve, performance will continue to increase.

**Initial Prompt**

A food caterer was told to prepare gourmet hot dogs for 36 guests. While most people would only eat one hotdog, he prepared enough for half of the guests to be able to have two hotdogs. However, 40 guests showed up, and everyone wanted a second hotdog. How many guests did not get a second hotdog?

**DebUnc: Debate with Uncertainty**

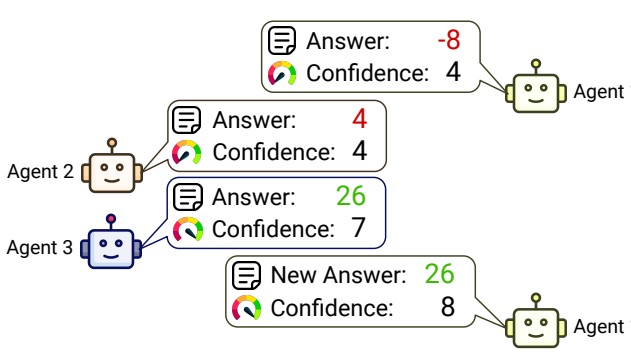

Figure 1: Illustration of a three-agent mathematical debate. Agent 1 initially provides an incorrect response, but corrects itself after seeing the responses and confidence levels from the other agents. Each agent uses a large language model (LLM) to generate text responses and assesses its confidence using an uncertainty metric. The responses and confidence information is shared among the agents, enabling them to decide whom to trust when responses differ. Correct answers are marked in green, while incorrect ones are shown in red.

## 1 Introduction

Large language models (LLMs) have shown impressive performance in various fields, including law, academia, and coding (OpenAI, 2024). To handle more complex tasks, LLM-powered agents have been developed. These agents observe their environment and take actions, such as communicating with other agents, using tools, or performing reasoning (Wu et al., 2023). The potential of LLM agents increases when multiple agents collaborate. One form of this is multi-agent debate, where agents propose and debate solutions to a problem with other agents. Multi-agent debates have been shown to improve the reasoning and accuracy of LLMs (Du et al., 2023).

The factual accuracy of LLMs is crucial for their utility in many real-world applications (Rawte et al., 2023). For example, an LLM tutor providing incorrect information could harm a student's learning, and an LLM customer service agent giving incorrect advice could mislead a customer. In fields such as healthcare, journalism, or financial services, the consequences of LLM errors can be even more severe, leading to financial losses or health risks. To address this, multi-agent debate systems have been proposed. These systems enable multiple agents to generate diverse responses to a problem, discuss and critique each other's answers, and ultimately converge on a final solution (Liang et al., 2023). Ideally, if some agents provide incorrect answers initially, the debate process helps them recognize and correct their errors.

In practice, while agents often confidently agree on the same final answer, that answer is not always correct. This can be attributed to flawed communication between agents (Du et al., 2023). LLM agents typically communicate through text and respond with a highly confident tone, regardless of of the accuracy of their answers. This creates a significant challenge in multi-agent systems, as a confidently incorrect response from one agent can mislead others, causing all agents to converge on an incorrect conclusion. By contrast, people often use qualifiers like "I am sure that..." or "I am not sure, but...," which, though imperfect, provide cues that help others gauge the reliability of the information. Since the tone of an LLM's response is not a reliable indicator of its accuracy, researchers have developed uncertainty metrics to provide a more objective measure of the model's confidence.

Building on these insights, we present DebUnc, a novel multi-agent debate framework that integrates multi-agent **Deb**ates with model **Unc**ertainty metrics. After each round of debate, we measure each agent's uncertainty with an uncertainty metric. In the following round, both the agents' responses and uncertainties are shared with the other agents. We explore two methods for conveying agent uncertainty: (1) incorporating the uncertainty directly into the textual prompt alongside the agent responses, as shown in Figure 2, and (2) adjusting the LLM's attention towards agents' responses based on their uncertainty, as depicted in Figure 3. We extensively evaluate DebUnc across multiple LLMs, benchmarks, and uncertainty metrics, analyzing the results of each uncertainty metric and method of uncertainty communication.

Our key contributions are outlined as follows:

- We introduce DebUnc, a framework that quantifies and communicates LLM agent uncertainty in multi-agent debates.
- We adapt the LLM attention mechanism to adjust token weights based on confidence levels and also explore the use of textual prompts to communicate confidence.
- We evaluate DebUnc across multiple LLMs, benchmarks, and uncertainty metrics, and find that the attention-scaling methods consistently outperforms unmodified debates.
- We offer insights into how performance will be impacted as uncertainty metrics improve.

## 2 RELATED WORK

LLMs are known for their overconfidence and their tendency to provide responses to any user query, regardless of their certainty. This often leads to the generation of factual inaccuracies, known as hallucinations, where the information provided by the model is incorrect or unsupported by the data on which it was trained (Liang et al., 2024; Yadkori et al., 2024; Duan et al., 2024; Yao et al., 2023; Aichberger et al., 2024). Ensuring factual accuracy is crucial for building trust in LLM-based systems and expanding their use in real-world applications. As a result, there has been a surge in research focused understanding the mechanisms behind hallucinations and developing strategies to mitigate them (Ji et al., 2023; McDonald et al., 2024; Liu et al., 2023).

### 2.1 UNCERTAINTY IN LLMS

Some current research efforts to mitigate hallucinations focus on measuring the model's uncertainty and enhancing their self-awareness (Kadavath et al., 2022; Amayuelas et al., 2023; Yin et al., 2023). If we could accurately measure a model's confidence, users would have clearer guidance on when to trust its output (Lin et al., 2022a; Xu et al., 2024), and language agents could better determine when to access external resources (Han et al., 2024).

However, accurately measuring model uncertainty remains an open problem, and ongoing research is exploring new uncertainty metrics. We identify three primary approaches to this problem:

**Token Probability-Based Uncertainty Metrics** evaluate uncertainty based on the probabilities assigned to each token generated by the model. High token probabilities (close to 1) indicate strong model confidence, whereas lower token probabilities suggest a spread of probability across several tokens, signifying uncertainty about the correct choice. Prominent methods in this category include Mean Token Entropy, Perplexity (Fomicheva et al., 2020), SAR (Duan et al., 2023), RDE (Vazhentsev et al., 2023), and Claim-Conditioned Probability (Fadeeva et al., 2024).

**LLM-Generated Uncertainty Metrics** involve the model explicitly expressing its uncertainty in its response. Lin et al. (2022a) explored this by fine-tuning GPT-3 (Brown et al., 2020) to provide both an answer and a corresponding confidence level. Alternatively, the model can be prompted to express its uncertainty without explicit training. Tian et al. (2023) found that this approach can can outperform token-probability-based methods when applied to LLMs fine-tuned with reinforcement learning from human feedback (Christiano et al., 2017). However, Xiong et al. (2023) report lower performance compared to token probability-based methods on GPT-3.

**Sampling-Based Uncertainty Metrics** assess uncertainty by generating multiple outputs through sampling, analyzing the distribution of meanings across the outputs. A consistent meaning across multiple samples suggests high confidence, while variations indicate lower confidence. This approach can identify different sequences that convey the same meaning, which token-probability-based metrics do not account for. However, the need for multiple generations makes these methods more resource-intensive than the others described. Examples include Semantic Entropy (Kuhn et al., 2023), LUQ (Zhang et al., 2024), and other metrics that evaluate meaning diversity (Lin et al., 2023).

## 2.2 MULTI-AGENT DEBATE

With the increased accessibility of LLMs and improvements in their inference times, LLMs are being integrated into more complex systems as autonomous agents (Wu et al., 2023; Li et al., 2023; Hong et al., 2023). A critical component of these agent-based systems is the collaboration mechanism, where models engage in debate with one another. These mechanisms are currently being studied (Zhang et al., 2023) and have been shown to foster more divergent thinking (Liang et al., 2023), enhance reasoning and factual accuracy (Du et al., 2023), and lead to more reliable evaluations (Chan et al., 2023). Through discussions, the LLMs can refine their outputs, ultimately achieving higher levels of agreement and producing more factually accurate text (Sun et al., 2024; Feng et al., 2024).

Pham et al. (2023) recognized that text is not be the most effective communication mechanism for LLM agents, since information is lost during the token sampling process, and demonstrated how LLMs can communicate through embeddings. ReConcile Chen et al. (2023) explored the integration of agent confidence in multi-agent debates, relying on LLM agents to self-report their confidence, which was communicated to other agents through prompts. Building on these ideas, we employ uncertainty metrics to estimate agent confidence and explore both prompting and attention scaling to convey this confidence. We found these enhancements to significantly improve multi-agent debate performance.

## 3 METHOD

In human debates, it is often possible to gauge someone's expertise on a subject by observing the fluency of their responses, their body language, and other cues. This helps in identifying whose arguments to consider more seriously when there are conflicting opinions. On the other hand, in multi-agent LLM debates, agents frequently generate inaccurate responses that sound confident, which can mislead other agents and result in a consensus on an incorrect response (Du et al., 2023). Our goal is to advise agents on which other agents' opinions to prioritize based on their confidence levels.

Our modified debate pipeline, depicted in Figure 2, operates as follows: in each round of debate, every agent generates a response, and its uncertainty is estimated. In the next round, the responses and uncertainties from each agent are shared with every other agent. We test three uncertainty metrics and three approaches to communicate agent uncertainty.

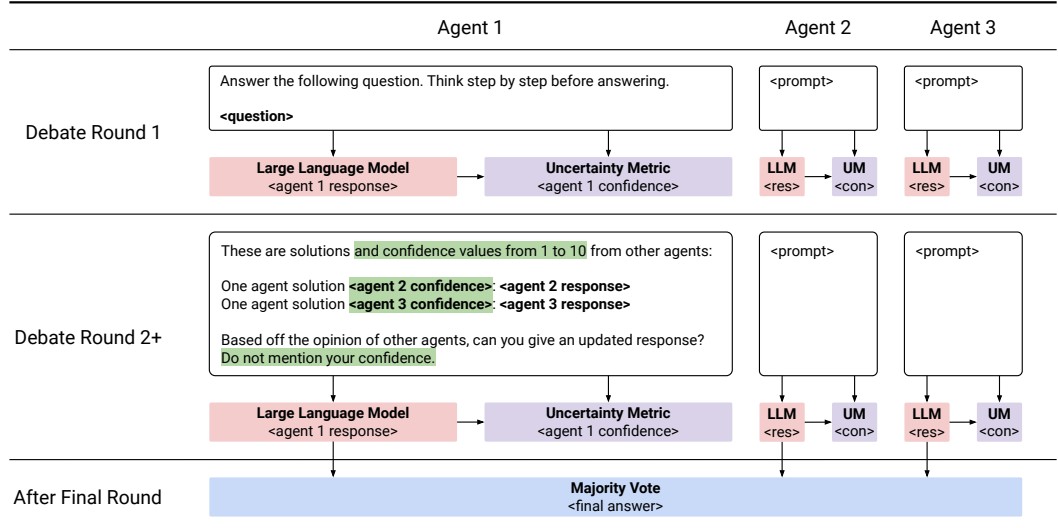

Figure 2: Illustration of the modified multi-agent debate involving three agents. In the first round, each agent independently generates a response to the question, which is evaluated for confidence using an uncertainty metric. The prompt for following rounds includes the responses from other agents in the previous round. Sections of the prompt highlighted in green are used only with the Confidence in Prompt method. Each agent retains access to its complete chat history throughout the debate. After the final round, a majority vote determines the final answer.

## 3.1 UNCERTAINTY METRICS

Uncertainty metrics assess an LLM's confidence in its responses: high uncertainty implies low confidence and potential unreliability, while low uncertainty suggests greater reliability. These metrics generally fall into three categories: Token Probability-Based, LLM-Generated, and Sampling-Based methods. For more details, see Uncertainty in LLMs

In our experiments, we focus on token probability-based metrics due to their efficiency and flexibility, requiring only a single generation and functioning independently of the model's ability to express uncertainty. We specifically chose Mean Token Entropy (Fomicheva et al., 2020) for its simplicity and TokenSAR (Duan et al., 2023), which accounts for the fact that some tokens contribute more to a sequence's meaning than others. We utilize the implementations from LM-Polygraph, a framework with implementations for many uncertainty metrics (Fadeeva et al., 2023). Lastly, to evaluate the potential of future uncertainty metrics, we include a third "oracle" uncertainty metric in our analysis.

**Mean Token Entropy.** One of the simplest and most efficient uncertainty to metrics to compute is Mean Token Entropy (Fomicheva et al., 2020). It is the average entropy across all tokens generated, with the entropy of a single token $X$ defined as:

$$H(X) = -\sum_{x \in V} p(x) \log p(x)$$

Here, $V$ denotes the model's vocabulary. Entropy is maximized when $p(x)$ is uniform over all tokens in the vocabulary, indicating maximum uncertainty. It is minimized when one token has a probability of 1 and all other tokens have a probability of 0, indicating complete certainty in the selected token.

**TokenSAR.** Duan et al. (2023) recognized that some tokens contribute more to a text's meaning than others and proposed TokenSAR, an uncertainty metric that accounts for this. It is defined as the weighted average of the negative log probabilities for each generated token, where the weights are the relevance scores of the tokens:

$$\text{TokenSAR} = \sum_i^N -\log p(t_i) R(t_i)$$

Here, $N$ is the number of tokens generated, $t_i$ is the $i$-th token, and $R(t_i)$ is the relevance of token $t_i$. To compute each token's relevance, RoBERTa-large (Liu et al., 2019) must be run $N$ times in total. This is more computationally expensive than calculating mean token entropy, but still far less costly than metrics requiring multiple generations.

**Oracle.**   While the uncertainty metrics discussed provide valuable insights into agent uncertainty, they are not without limitations, and future advancements will likely lead to more accurate metrics. To assess the potential effectiveness of our methods with improved uncertainty metrics, we include an "Oracle" metric, which simulates an ideal uncertainty metric. This metric yields low uncertainty when the agent is correct and high uncertainty when the agent is incorrect, and is defined as follows:

$$\text{uncertainty} = \begin{cases} 0 & \text{if the response is correct} \\ \infty & \text{if the response is incorrect} \end{cases}$$

In practice, using 0 and $\infty$ could cause issues with our uncertainty communication methods, so we detail exactly how this metric is applied in the following subsection. It is also important to note that this metric requires knowledge of the ground truth answer, making it impractical for real-world use. Instead, it serves to evaluate the effectiveness of our uncertainty communication methods independently of the performance of the uncertainty metrics themselves, and allows us to anticipate how improvements in uncertainty metrics could affect debate performance.

### 3.2   UNCERTAINTY COMMUNICATION

After computing the uncertainty of each agent, we explore multiple methods to incorporate these uncertainties into the following debate round.

**Confidence in Prompt.**   One approach is to include the uncertainties directly in the text prompt for the next debate round, as shown in Figure 2. Mean Token Entropy and TokenSAR yield non-negative uncertainties. For Mean Token Entropy, the range of uncertainties depends on the model's vocabulary size, while for TokenSAR, the maximum uncertainty is unbounded. Therefore, the exact uncertainty values are less informative than the relative differences in uncertainty between agents.

Rather than expressing their uncertainty as an unbounded non-negative number, humans often express their confidence on 1 to 10 scale, which more interpretable. Since LLMs are trained on human data, they may exhibit the same preference. As a result, we convert the uncertainties into confidence values. Given a list of uncertainties $u$ for $n$ agents, where $u_i$ is the uncertainty of agent $i$, we first invert them to obtain raw confidence values $r_i = \frac{1}{u_i}$. We then scale these values such that the average confidence $s_i$ of all agents is 5:

$$s_i = \frac{r_i}{\sum_{j=1}^{n} r_j} \cdot (5n - 1) + \frac{1}{n}$$

Finally, we clamp the confidence levels to the range of 1 to 10 and round to the nearest integer.

When using the Oracle uncertainty metric, we set the confidence to 1 if the agent was incorrect and to 10 if the agent was correct.

**Attention Scaling.**   As an alternative to including confidence levels in the prompt, we can modify the LLM's token-generation process to account for each agent's confidence. Many LLMs use Transformer decoder layers that generate an embedding for the last token and use this embedding to predict the next token (Radford et al., 2018). This embedding is determined by the attention mechanism, which creates "query," "key," and "value" vectors for each token.

The similarity between the "query" vector of the last token and the "key" vector of each token is used to compute a weight for every token. These weights are normalized with a softmax function to ensure they sum to 1, and are used to create the output embedding, which is the weighted sum of the value vectors of each token (Vaswani et al., 2017). The weight of each token determines its influence on the next token generated. By modifying these weights, we can adjust the model's focus on each token in the input.

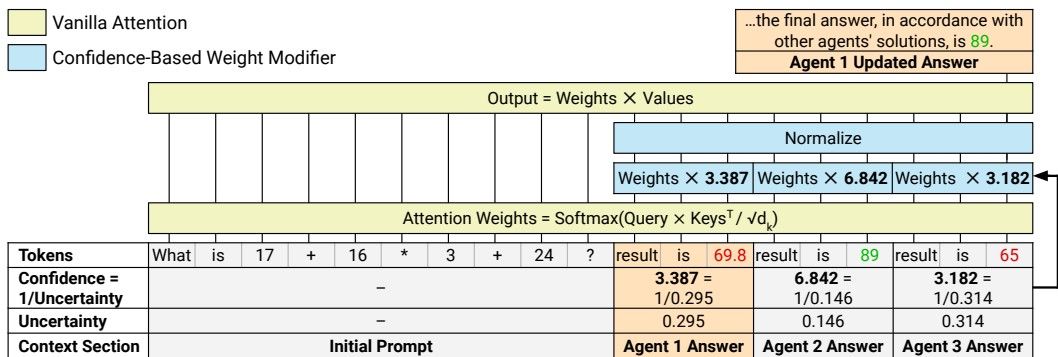

Figure 3: Illustration of the Attention-All method from the perspective of Agent 1. As the second debate round begins, the model's context includes the initial prompt and each agent's responses. Agent 2 provided a correct response with lower uncertainty than Agents 1 and 3, who responded incorrectly. Because Agent 2 had a lower uncertainty, the attention weights for tokens constituting Agent 2's response will be increased, while those for tokens from Agent 1 and Agent 3's responses will be decreased. This led Agent 1 to switch to the correct answer.

In multi-agent debates, this allows us to shift the model's focus towards the responses from more confident agents. After each debate round, we have responses from each agent. In the next round, each agent's prompt will include these responses. We also compute the uncertainty of each agent using an uncertainty metric.

In the next round, as the LLM generates its response, it computes the normalized attention weights for each preceding token. We multiply the weight of every token from agent $j$ by the inverse of agent $j$'s uncertainty uncertainty when using Mean Token Entropy or TokenSAR. When using the Oracle metric, to avoid divide-by-zero errors, we set the multiplier to $10^{-5}$ if the agent was incorrect and 1 if the agent was correct.

Formally, the attention weight for token i $a_i$ is:

$$a_i = \begin{cases} w_i \cdot m_j & \text{if } i \in t_j, \text{ for any agent } j \\ w_i & \text{otherwise} \end{cases}$$

Here, $t_j$ is the set of token indices from agent $j$. We then normalize the scaled attention weights to ensure that the sum of all token weights equals 1, while leaving the weights of other tokens unchanged. The final weight $f_i$ for every token $i$ is calculated as follows:

$$f_i = \begin{cases} a_i \cdot \frac{\sum_{j=1}^n \sum_{k \in t_j} w_k}{\sum_{j=1}^n \sum_{k \in t_j} a_k} & \text{if } i \in t_j, \\ & \text{for any agent } j \\ a_i & \text{otherwise} \end{cases}$$

We only apply attention scaling to the responses from the previous round. For instance, in a three-round debate, attention would be rescaled for the responses from the first round during the second round, and for the responses from the second round during the third round. In the third round of debate, attention would not be rescaled to the first-round responses. Additionally, in order to prevent divide-by-zero errors during normalization, attention is not scaled when computing the embeddings for tokens within the prompt; it is only scaled when generating new tokens.

We explore two variants of attention scaling:

• **Attention-Others**, where agent $i$ only rescales attention to other agents' response tokens $t_j \mid j \neq i$

• **Attention-All**, where agent $i$ rescales attention to other agents and itself, illustrated in Figure 3

| Metric | Method | MMLU-0 | MMLU-5 | GSM8k | Arith. | Truth. | Average |
|--------|--------|--------|--------|-------|--------|--------|---------|
| N/A | Standard | 0.520 | 0.544 | 0.512 | 0.478 | 0.604 | 0.532 +0.0% |
| Entropy | Prompt | 0.522 | 0.546 | **0.536** | 0.482 | 0.602 | 0.538 +1.1% |
| | Attn-Others | **0.540** | 0.566 | 0.488 | **0.518** | **0.608** | 0.544 +2.3% |
| | Attn-All | 0.526 | **0.570** | **0.536** | 0.518 | 0.604 | **0.551** +3.6% |
| TokenSAR | Prompt | **0.538** | 0.548 | 0.504 | 0.464 | 0.616 | 0.534 +0.4% |
| | Attn-Others | 0.526 | **0.560** | 0.496 | **0.500** | 0.626 | 0.542 +1.9% |
| | Attn-All | 0.532 | 0.552 | **0.528** | 0.500 | 0.610 | **0.544** +2.3% |
| Oracle | Prompt | 0.562 | 0.560 | 0.548 | 0.542 | 0.626 | 0.568 +6.8% |
| | Attn-Others | 0.606 | 0.666 | 0.640 | 0.654 | 0.630 | 0.639 +20.1% |
| | Attn-All | **0.618** | **0.684** | **0.656** | **0.732** | **0.648** | **0.668** +25.6% |

Table 1: Accuracy comparison across various benchmarks using different uncertainty estimators and methods with Mistral-7B. 'MMLU-0' denotes zero-shot performance on MMLU, while 'MMLU-5' represents 5-shot performance. The other benchmarks used zero-shot prompting. The 'Average' column shows the average performance for all tests and the % increase over the standard debate.

## 4 EXPERIMENT DESIGN

To evaluate these methods, an open-source LLM is required, as implementing the attention scaling requires modifications to the model source code. Additionally, the uncertainty metrics used rely on token probabilities, which may not be readily available from closed-source models. We ran all of the experiments on Mistral-7B-Instruct-v0.2 (Jiang et al., 2023), and re-evaluated the most insightful ones on Llama-3-8B-Instruct (AI@Meta, 2024) to verify that our methods work across different models. Tokens were sampled with a temperature of 1 to ensure variability in the responses. The methods were evaluated on multiple benchmarks:

1. **MMLU** (Hendrycks et al., 2021): A dataset of multiple-choice questions across various subjects.

2. **GSM8k** (Cobbe et al., 2021): A dataset of free-response grade school math problems.

3. **TruthfulQA** (Lin et al., 2022b): A multiple-choice dataset testing the model's susceptibility to common misconceptions.

4. **Arithmetic**: An randomly generated set of arithmetic problems in the form $a + b \cdot c + d$ for the Mistral model and $a + b \cdot c + d - e \cdot f$ for the Llama model, where $0 \leq a, b, c, d < 30$.

For MMLU on Mistral-7B, we tested both zero-shot and 5-shot prompting, using examples from the original MMLU repository's[1] dev set. For the remaining benchmarks, we used only zero-shot prompting. The exact prompts used are shown in Appendix C.

Instead of evaluating on the full datasets, which would be too expensive, we sampled 100 questions from each. We evaluated every combination of uncertainty metric and uncertainty incorporation method on these samples five times, and report the average accuracy across the five runs.

The debates involved three agents and three rounds, with all agents using the same LLM. In the first round, each agent answered the question individually. In the following rounds, each agent was given other agents' responses, and used this information to refine their answers. A full example debate is shown in Appendix A. The final answer was determined by a majority vote after the third round. This workflow is illustrated in Figure 2.

## 5 RESULTS

In this section, we first analyze the effectiveness of each uncertainty incorporation method, and then analyze the effectiveness of the uncertainty metrics.

---

[1]https://github.com/hendrycks/test

| Metric | Method | MMLU-0 | GSM8k | TruthfulQA | Arithmetic | Average |
|--------|--------|--------|-------|------------|------------|---------|
| N/A | Standard | 0.654 | 0.812 | 0.518 | 0.520 | 0.634 +0.0% |
| Entropy | Prompt | 0.614 | **0.840** | 0.540 | 0.528 | 0.636 +0.4% |
| | Attention-Others | 0.638 | 0.810 | **0.562** | 0.526 | **0.639** +0.9% |
| | Attention-All | **0.658** | 0.808 | 0.556 | **0.530** | 0.638 +0.4% |
| Oracle | Prompt | 0.668 | 0.872 | 0.584 | 0.546 | 0.668 +5.5% |
| | Attention-Others | **0.784** | 0.898 | 0.674 | **0.556** | **0.728** +14.9% |
| | Attention-All | 0.754 | **0.900** | **0.682** | 0.556 | **0.728** +15.0% |

Table 2: Accuracy comparison across various benchmarks using different uncertainty estimators and methods with Llama-3-8B. Zero-shot prompting was used for all benchmarks. The 'Average' column shows the average performance for all tests and the % increase over the standard debate.

## 5.1 UNCERTAINTY INCORPORATION METHODS

Table 1 presents a comparison of the results obtained with Mistral-7B using different combinations of uncertainty metrics (Mean Token Entropy, TokenSAR, and Oracle) and methods (Confidence in Prompt, Attention-Others, and Attention-All). As a baseline, the performance of a standard 3-agent, 3-round debate without any uncertainty metrics is also shown. Overall, Attention-All was the top-performing method, achieving the highest average accuracy across all three uncertainty metrics. It was the only method that consistently matched or exceeded the performance of the standard multi-agent debate on all benchmarks. Table 2 presents the results using Llama-3-8B. The findings show that with the Oracle uncertainty metric, the attention scaling methods significantly outperformed confidence communication via prompting. However, when using mean token entropy, the performance gains were minimal, suggesting that mean token entropy may not be as effective on Llama-3-8B as it was on Mistral-7B.

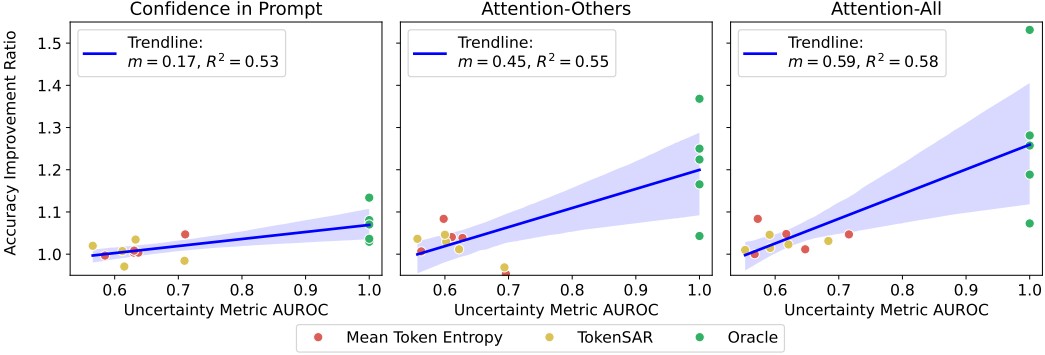

Figure 4: Plots of the ratio of accuracy improvement over a standard debate against the uncertainty metric AUROC for all Mistral-7B experiments. A higher AUROC indicates better metric performance. Each point represents the results on one of the benchmarks (MMLU-0, MMLU-5, GSM8k, Arithmetic, and TruthfulQA). The plots are organized by the method of uncertainty incorporation method (as titled) and the uncertainty metric used (color-coded). The trendlines show that attention-based methods, especially Attention-All, lead to more substantial performance gains as AUROC increases compared to methods that incorporate confidence directly into the prompt.

As shown in Figure 4, Attention-All demonstrates the most significant accuracy improvements as the AUROC of the uncertainty metric increases, with a slope of 0.59 compared to 0.45 for Attention-Others and 0.17 for Confidence in Prompt. The accuracy improvement ratio compares the method's accuracy to the accuracy observed in a standard debate. AUROC, the area under the receiver operating characteristic curve, represents the probability that a correct answer is assigned a lower uncertainty than an incorrect one. A random uncertainty metric would have an AUROC of 0.5, while a perfect one would have an AUROC of 1.

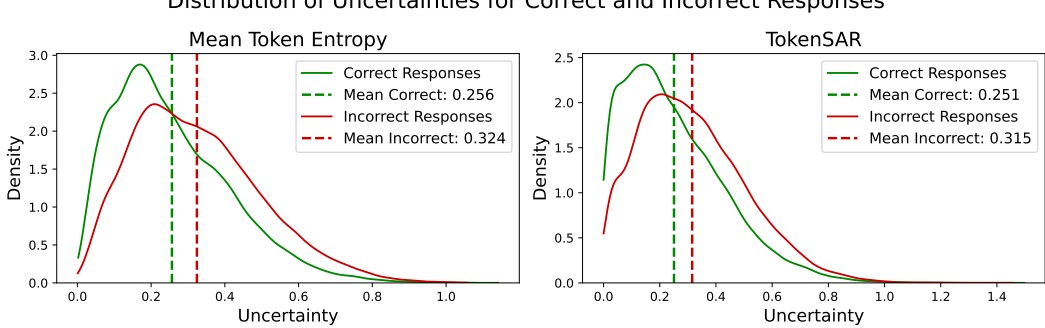

Figure 5: Distribution of uncertainties for correct and incorrect answers across all Mistral-7B experiments, as measured by the uncertainty metrics Mean Token Entropy and TokenSAR. Generally, correct answers exhibit lower uncertainties than incorrect ones, indicating that although not perfect, uncertainty metrics are useful for distinguishing between accurate responses and those where the agent may be hallucinating.

## 5.2 UNCERTAINTY METRICS

The best-performing uncertainty metric was the Oracle metric. Mean Token Entropy ranked next, with debates using it consistently achieving higher average accuracies than debates using TokenSAR, as shown in Table 1. Mean Token Entropy achieved an average AUROC across all experiments of 0.627, compared to 0.617 for TokenSAR. Full AUROC results are shown in Appendix B.

To further analyze the uncertainty metrics and understand their impact, Figure 5 presents the distributions of uncertainties for responses containing correct answers versus incorrect answers across all benchmarks and uncertainty incorporation methods. The left plot illustrates the uncertainties when using Mean Token Entropy, while the right plot depicts uncertainties when using TokenSAR. As shown in Table 1, Mean Token Entropy achieved slightly higher accuracy than TokenSAR on average. Examining the average uncertainties for correct and incorrect responses, Mean Token Entropy has a ratio of average correct to average incorrect of 1.264, compared to 1.258 for TokenSAR. This suggests that Mean Token Entropy slightly outperforms TokenSAR in differentiating between correct and incorrect responses, while also being less computationally demanding. However, the performance difference is minimal and possibly falls within the margin of error.

## 6 CONCLUSION

We introduce DebUnc, a framework that integrates confidence estimations into multi-agent debates. By incorporating confidence levels, agents can assess the reliability of their peers and better determine whom to trust when responses differ. We use uncertainty metrics to estimate the LLM's confidence based on the probability distribution generated for each token. To convey confidence, we adjusted the LLM's attention mechanism to scale token weights according to confidence levels and also experimented with using textual prompts. Our results show that attention scaling is more effective, demonstrating that information can be delivered to LLMs through methods beyond traditional text prompts. Attention scaling can be applied in any scenario where certain parts of the input need to be prioritized over others, and we encourage further exploration of this technique.

## 7 LIMITATIONS

One limitation of DebUnc is its reliance on open-source LLMs, as attention scaling requires code modifications, and token probabilities may not be accessible from proprietary models. Additionally, attention scaling is sensitive to the order of agent responses within the prompt. For example, if agent 2's response precedes agent 3's, unidirectional attention allows agent 2 to influence agent 3's tokens, potentially leaking information from agent 2 even if agent 3 is more confident.

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

## A EXAMPLE DEBATE

This is a full debate on zero-shot MMLU using Mistral-7B as the LLM. It uses the Confidence in Prompt method for incorporating uncertainty with the Oracle uncertainty metric. For conciseness, the user prompts for rounds 2 and 3 use "<agent $i$ response>" to reference the response from agent $i$ in the previous round.

| **Agent 1** | **Agent 2** | **Agent 3** |
|---|---|---|
| *User, Round 1* | | |

Answer the following multiple choice question. The last line of your response should be of the following format: 'Answer: $LETTER' (without quotes) where LETTER is one of ABCD. Think step by step before answering.

Which of these combinations of clinical features is most suggestive of mixed mitral valve disease with a predominance of mitral regurgitation?
A. Displaced apex beat; soft first heart sound; pan-systolic murmur; short mid-diastolic murmur
B. Irregularly, irregular pulse; displaced apex beat; ejection systolic murmur; short mid-diastolic murmur
C. Displaced apex beat; normal first heart sound; pan-systolic murmur; long mid-diastolic murmur
D. Tapping apex beat; loud first heart sound; pan-systolic murmur; long mid-diastolic murmur

| | *Assistant, Round 1* | |
|---|---|---|

| **Agent 1** | **Agent 2** | **Agent 3** |
|---|---|---|
| Based on the given clinical features, option C: Displaced apex beat; normal first heart sound; pan-systolic murmur; long mid-diastolic murmur is least suggestive of mixed mitral valve disease with a predominance of mitral regurgitation.

The features most suggestive of mixed mitral valve disease with a predominance of mitral regurgitation are:

1. Displaced apex beat: This can occur due to chronic volume overload or dilatation of the left ventricle from mitral regurgitation.
2. Soft or absent first heart sound: This can be due to the decreased force of mitral valve closure during diastole.
3. Pan-systolic murmur: This is typically heard in mitral regurgitation due to the continuous regurgitant jet throughout the cardiac cycle.
4. Long mid-diastolic murmur: This can occur due to the continued mitral regurgitant jet during diastole.

Therefore, the correct answer is:

A. Displaced apex beat; soft first heart sound; pan-systolic murmur; long mid-diastolic murmur

Answer: A | Based on the given clinical features, the combination most suggestive of mixed mitral valve disease with a predominance of mitral regurgitation would be:

C. Displaced apex beat; normal first heart sound; pan-systolic murmur; long mid-diastolic murmur

Explanation:
The presence of a displaced apex beat and a pan-systolic murmur points towards mitral valve disease. However, the duration and timing of the additional diastolic murmur helps distinguish the underlying etiology. A long mid-diastolic murmur is more suggestive of functional mitral regurgitation, which is a component of mixed mitral valve disease with a predominance of mitral regurgitation. Therefore, option C is the correct answer.

Answer: C | The combination of a displaced apex beat, a pan-systolic murmur, and a long mid-diastolic murmur is most suggestive of mitral stenosis. Thus, the answer to the question is D. however, we were asked for a combination that is most suggestive of mixed mitral valve disease with a predominance of mitral regurgitation. With this in mind, the correct answer is: C. Displaced apex beat; normal first heart sound; pan-systolic murmur; long mid-diastolic murmur

Explanation: Mixed mitral valve disease refers to a condition in which both mitral regurgitation and mitral stenosis are present. The presence of a predominantly regurgitant jet suggests a more advanced state of the disease. A displaced apex beat can be seen in conditions of volume overload, such as mitral regurgitation, and is less common in mitral stenosis. A normal first heart sound and a long mid-diastolic murmur are more likely to be heard in mitral regurgitation than mitral stenosis, as they represent the opening and closing of the mitral valve during diastole. Therefore, option C, with a displaced apex beat, normal first heart sound, and a long mid-diastolic murmur, is the most suggestive of mixed mitral valve disease with a predominance of mitral regurgitation.

Answer: C. |

| Agent 1 (cont.) | Agent 2 (cont.) | Agent 3 (cont.) |
|---|---|---|

*User, Round 2*

| These are solutions and confidence values from 1 to 10 (higher means more confident) to the problem from other agents: | These are solutions and confidence values from 1 to 10 (higher means more confident) to the problem from other agents: | These are solutions and confidence values from 1 to 10 (higher means more confident) to the problem from other agents: |
|---|---|---|
| One agent solution (confidence level is 1): "'<agent 2 response>"' | One agent solution (confidence level is 10): "'<agent 1 response>"' | One agent solution (confidence level is 10): "'<agent 1 response>"' |
| One agent solution (confidence level is 1): "'<agent 3 response>"' | One agent solution (confidence level is 1): "'<agent 3 response>"' | One agent solution (confidence level is 1): "'<agent 2 response>"' |
| Based off the opinion of other agents, can you give an updated response? Do not mention your confidence. Think step by step before answering. The last line of your response should be of the following format: 'Answer: $LETTER' (without quotes) where LETTER is one of ABCD. | Based off the opinion of other agents, can you give an updated response? Do not mention your confidence. Think step by step before answering. The last line of your response should be of the following format: 'Answer: $LETTER' (without quotes) where LETTER is one of ABCD. | Based off the opinion of other agents, can you give an updated response? Do not mention your confidence. Think step by step before answering. The last line of your response should be of the following format: 'Answer: $LETTER' (without quotes) where LETTER is one of ABCD. |

*Assistant, Round 2*

| Based on the information provided by the other agents, the combination most suggestive of mixed mitral valve disease with a predominance of mitral regurgitation is: | Based on the clinical features provided and the consensus of other agents, option C: Displaced apex beat; normal first heart sound; pan-systolic murmur; long mid-diastolic murmur is the most suggestive of mixed mitral valve disease with a predominance of mitral regurgitation. | Based on the discussions among different agents, the most suggestive combination of clinical features for mixed mitral valve disease with a predominance of mitral regurgitation is: |
|---|---|---|
| A. Displaced apex beat; soft first heart sound; pan-systolic murmur; long mid-diastolic murmur. | Answer: C. | A. Displaced apex beat; soft first heart sound; pan-systolic murmur; long mid-diastolic murmur |
| The presence of a displaced apex beat, soft first heart sound, and prolonged pan-systolic murmur indicate mitral valve disease. The duration and timing of the diastolic murmur can be used to distinguish between mitral stenosis and mitral regurgitation. A long, holodiastolic murmur is more commonly heard in mitral regurgitation than mitral stenosis. Therefore, the presence of a long mid-diastolic murmur with a displaced apex beat and soft first heart sound suggests mixed mitral valve disease with a predominance of mitral regurgitation. | | Explanation: Mixed mitral valve disease is characterized by the co-existence of mitral stenosis and mitral regurgitation. The presence of a displaced apex beat, soft first heart sound, pan-systolic murmur, and long mid-diastolic murmur can suggest a diagnosis of mixed mitral valve disease with a predominance of mitral regurgitation. The displaced apex beat can occur due to chronic volume overload or dilatation of the left ventricle, as seen in mitral regurgitation. The soft or absent first heart sound can be due to the decreased force of mitral valve closure in mitral regurgitation. The pan-systolic murmur is typically heard in mitral regurgitation due to the continuous regurgitant jet throughout the cardiac cycle. The long mid-diastolic murmur can occur due to the continued mitral regurgitant jet during diastole. Therefore, the combination of these clinical findings, A. Displaced apex beat; soft first heart sound; pan-systolic murmur; long mid-diastolic murmur, is most suggestive of mixed mitral valve disease with a predominance of mitral regurgitation. |
| Answer: A. | | Answer: A. |

| Agent 1 (cont.) | Agent 2 (cont.) | Agent 3 (cont.) |
|---|---|---|
| *User, Round 3* | | |
| These are solutions and confidence values from 1 to 10 (higher means more confident) to the problem from other agents: | These are solutions and confidence values from 1 to 10 (higher means more confident) to the problem from other agents: | These are solutions and confidence values from 1 to 10 (higher means more confident) to the problem from other agents: |
| One agent solution (confidence level is 1): "'<agent 2 response>'" | One agent solution (confidence level is 10): "'<agent 1 response>'" | One agent solution (confidence level is 10): "'<agent 1 response>'" |
| One agent solution (confidence level is 10): "'<agent 3 response>'" | One agent solution (confidence level is 10): "'<agent 3 response>'" | One agent solution (confidence level is 1): "'<agent 2 response>'" |
| Based off the opinion of other agents, can you give an updated response? Do not mention your confidence. Think step by step before answering. The last line of your response should be of the following format: 'Answer: $LETTER' (without quotes) where LETTER is one of ABCD. | Based off the opinion of other agents, can you give an updated response? Do not mention your confidence. Think step by step before answering. The last line of your response should be of the following format: 'Answer: $LETTER' (without quotes) where LETTER is one of ABCD. | Based off the opinion of other agents, can you give an updated response? Do not mention your confidence. Think step by step before answering. The last line of your response should be of the following format: 'Answer: $LETTER' (without quotes) where LETTER is one of ABCD. |
| *Assistant, Round 3* | | |
| Based on the discussions among different agents, the combination of clinical features most suggestive of mixed mitral valve disease with a predominance of mitral regurgitation is:

A. Displaced apex beat; soft first heart sound; pan-systolic murmur; long mid-diastolic murmur.

Explanation:
Mixed mitral valve disease is defined as the presence of both mitral stenosis and mitral regurgitation. The clinical presentation of mixed mitral valve disease can vary depending on the severity and predominance of each component.

A displaced apex beat, soft first heart sound, pan-systolic murmur, and long mid-diastolic murmur are more indicative of mitral regurgitation than mitral stenosis. The displaced apex beat can result from the increased volume load in the left ventricle due to mitral regurgitation. A soft or absent first heart sound may be heard in mitral regurgitation due to the decreased force of mitral valve closure, while the pan-systolic murmur and long mid-diastolic murmur are characteristic features of mitral regurgitation.

Therefore, the combination of these clinical findings, A. Displaced apex beat; soft first heart sound; pan-systolic murmur; long mid-diastolic murmur, is most suggestive of mixed mitral valve disease with a predominance of mitral regurgitation.

Answer: A. | Based on the consensus of other agents, the combination most suggestive of mixed mitral valve disease with a predominance of mitral regurgitation is: A. Displaced apex beat; soft first heart sound; pan-systolic murmur; long mid-diastolic murmur.

Answer: A. | Based on the information provided by various agents and the consensus reached, the combination of clinical features most suggestive of mixed mitral valve disease with a predominance of mitral regurgitation is:

A. Displaced apex beat; soft first heart sound; pan-systolic murmur; long mid-diastolic murmur

Here's the explanation:
The presence of a displaced apex beat and a pan-systolic murmur suggest mitral valve disease. However, the duration and timing of the diastolic murmur can help distinguish between mitral stenosis and mitral regurgitation. Mitral regurgitation is more commonly associated with a long, holodiastolic or mid-diastolic murmur, while mitral stenosis typically has a short, mid-diastolic murmur. In this scenario, the long mid-diastolic murmur, along with the displaced apex beat and soft first heart sound, makes mixed mitral valve disease with a predominance of mitral regurgitation the most likely diagnosis. Therefore, the answer is:

Answer: A. |

## B  FULL UNCERTAINTY METRIC RESULTS

| Estimator | Method | MMLU-0 | MMLU-5 | GSM8k | Arithmetic | TruthfulQA | Avg. |
|---|---|---|---|---|---|---|---|
| Entropy | Prompt | 0.630 | 0.637 | 0.711 | 0.631 | 0.585 | 0.639 |
|  | Attn-Others | 0.628 | 0.611 | 0.696 | 0.598 | 0.563 | 0.619 |
|  | Attn-All | 0.647 | 0.618 | 0.716 | 0.573 | 0.568 | 0.624 |
| TokenSAR | Prompt | 0.633 | 0.612 | 0.710 | 0.615 | 0.566 | 0.627 |
|  | Attn-Others | 0.622 | 0.602 | 0.694 | 0.600 | 0.557 | 0.615 |
|  | Attn-All | 0.621 | 0.592 | 0.684 | 0.591 | 0.553 | 0.608 |
| Oracle | Prompt | 1.000 | 1.000 | 1.000 | 1.000 | 1.000 | 1.000 |
|  | Attn-Others | 1.000 | 1.000 | 1.000 | 1.000 | 1.000 | 1.000 |
|  | Attn-All | 1.000 | 1.000 | 1.000 | 1.000 | 1.000 | 1.000 |

Table 3: This table displays the uncertainty metric AUROC values for each experiment run with Mistral-7B. 'MMLU-0' denotes zero-shot prompting on MMLU, while 'MMLU-5' represents 5-shot prompting. The other benchmarks used zero-shot prompting. The 'Avg.' column shows the average AUROC for all tests. By definition, the Oracle metric achieved perfect AUROC scores. Among the remaining metrics, Mean Token Entropy delivered slightly better performance than TokenSAR despite being cheaper to run.

## C  PROMPTS

The table below displays the prompts used to facilitate the debates. Text in green is only included when using the Confidence in Prompt method, and text in blue is specific to each debate.

| Benchmark | Prompt Type | Prompt |
|---|---|---|
| MMLU | Initial, 0-shot | Answer the following multiple choice question. The last line of your response should be of the following format: 'Answer: $LETTER' (without quotes) where LETTER is one of ABCD. Think step by step before answering.

<question> |
| | Initial, 5-shot | Answer the following multiple choice question.

Examples:

<5 example questions and answers>

—

YOUR TASK

Answer the following question. Think step by step before answering. The last line of your response should be of the following format: 'Answer: $LETTER' (without quotes) where LETTER is one of ABCD.

<question> |
| | Debate | These are solutions and confidence values from 1 to 10 (higher means more confident) to the problem from other agents:

One agent solution (confidence level is $c$): <agent response>

One agent solution (confidence level is $c$): <agent response>

Based off the opinion of other agents, can you give an updated response? Do not mention your confidence. Think step by step before answering. The last line of your response should be of the following format: 'Answer: $LETTER' (without quotes) where LETTER is one of ABCD. |
| GSM8k | Initial | Answer the following math problem. The last line of your response should be of the following format: 'Answer: $INTEGER' (without quotes) where INTEGER is the integer answer. Think step by step before answering.

<question> |
| | Debate | These are solutions and confidence values from 1 to 10 (higher means more confident) to the problem from other agents:

One agent solution (confidence level is $c$): <agent response>

One agent solution (confidence level is $c$): <agent response>

Based off the opinion of other agents, can you provide an updated response? The original problem is:

<question>

Do not mention your confidence. The last line of your response should be of the following format: 'Answer: $INTEGER' (without quotes) where INTEGER is the integer answer. |

| Benchmark | Prompt Type | Prompt |
|---|---|---|
| Arithmetic | Initial | What is the result of \<a+b*c+d\>? State the final answer at the end of your response. |
| | Debate | These are solutions and confidence values from 1 to 10 (higher means more confident) to the problem from other agents:

One agent solution (confidence level is $c$): \<agent response\>

One agent solution (confidence level is $c$): \<agent response\>

Based off the opinion of other agents, can you provide an updated answer? Do not mention your confidence. State the final answer at the end of your response. |
| TruthfulQA | Initial | Answer the following multiple choice question:

\<question\>

Think step by step before answering.  The last line of your response should be of the following format: 'Answer: $LETTER' (without quotes) where LETTER is one of \<options\>. |
| | Debate | These are the selections and confidence values from 1 to 10 (higher means more confident) from other agents:

One agent solution (confidence level is $c$): \<agent response\>

One agent solution (confidence level is $c$): \<agent response\>

Can you double check that your response is correct?  Do not mention your confidence. The last line of your response should be of the following format: 'Answer: $LETTER' (without quotes) where LETTER is one of \<options\>. |

