# OpenReview forum: "DebUnc: Improving Large Language Model Agent Communication Via Uncertainty Metrics"
_ICLR.cc/2025/Conference — ICLR 2025 Conference Withdrawn Submission_

### Official Review · Reviewer_s5TV · 2024-10-18

**Soundness:** 4
**Presentation:** 3
**Contribution:** 2
**Rating:** 3
**Confidence:** 5

**Summary:**

One problem in multi-agent communication is that the uncertainty of utterances is not well captured. This paper directly compensate this by proposing an improvement of the vanilla communication scheme---predicting uncertainty from the answer and put that in the utterance as well. Similar to majority voting but with weights, at a higher level.

**Strengths:**

+ The proposed technique is well motivated.

**Weaknesses:**

+ How sure are we that the language model won't express their uncertainties via natural language if we prompt them well enough? I'm expecting this capability should be attainable with few-shot prompts.
+ A big confounder is that by tuning the hyperparameters of the uncertainty metrics, we actually find a predictor of the correctness. And this alone (instead of communication) is the real drive behind improved scores. A n ablation is needed for a simple weighted majority vote.
+ As mentioned in L141, Pham et al. and Chen et al. are other improvements made on Du et al. How does the proposed technique compare to them? (just asking, these can be argued to be contemporaneous)

**Questions:**

+ In Table 1 and 2, there is a risk that oracle could be interpreted as one of the proposed methods. It should be visually more separated.

---

### Official Review · Reviewer_EZeH · 2024-11-01

**Soundness:** 2
**Presentation:** 2
**Contribution:** 2
**Rating:** 3
**Confidence:** 4

**Summary:**

This paper introduces DebUnc, a framework that improves multi-agent LLM debates by incorporating uncertainty metrics to address overconfident incorrect responses. The framework communicates uncertainty either through text prompts or by adjusting the LLM's attention mechanism. Experiments across multiple benchmarks show that the attention-based approach consistently outperforms standard debates.

**Strengths:**

1. The paper presents its core idea in a clear and straightforward manner. The proposed solution of incorporating uncertainty metrics into multi-agent debates is simple.

2. The figures effectively communicate the key concepts.

**Weaknesses:**

1. The experimental comparisons are insufficient. The authors should have included basic baselines like Chain-of-Thought (CoT) and compared their method with prior work on multi-agent debates [1,2]. A simple baseline of having agents directly generate uncertainty in their responses is also missing. More importantly, since multi-agent debate is quite similar to self-consistency (both generate multiple answers), they should compare with CoT self-consistency using similar computation budgets. They could have also tried applying uncertainty metrics directly to self-consistency, which might be simpler than their proposed approach.

2. In the "Attention Scaling" section, many key notations ($w_i$, $m_j$, $f_i$) are just thrown in without proper definition. Some implementation choices, like only applying attention scaling to the previous round's responses, aren't explained or validated through ablation studies.

3. Several important implementation details are inadequately explained: the decision to "only apply attention scaling to the responses from the previous round" lacks justification, and no ablation studies to validate such design choices.

4. I'm also confused by Figure 4. How did authors get different data points for each method? Run with different random seeds? Also, the trend looks weak if we ignore the oracle metric (which I think we should, since it's completely impractical). And it's concerning that for "Attention-Others", the accuracy actually drops when AUROC increases to around 0.7.

5. The improvements are marginal and unconvincing. With proposed uncertainty metrics, the authors only get 1-3% improvement, and less than 1% for Llama 3 8B. Given how much more complex and computationally expensive their method is, these gains are hard to justify.

Overall, while the paper presents an interesting direction, the lack of comprehensive comparisons, unclear technical details, and marginal improvements make it difficult to assess the true value of the contribution.

[1] Yilun Du, Shuang Li, Antonio Torralba, Joshua B. Tenenbaum, and Igor Mordatch. Improving factuality and reasoning in language models through multiagent debate.

[2] Tian Liang, Zhiwei He, Wenxiang Jiao, Xing Wang, Yan Wang, Rui Wang, Yujiu Yang, Zhaopeng Tu, and Shuming Shi. Encouraging divergent thinking in large language models through multi-agent debate.

**Questions:**

See above weaknesses

---

### Official Review · Reviewer_XH6i · 2024-11-04

**Soundness:** 2
**Presentation:** 3
**Contribution:** 2
**Rating:** 3
**Confidence:** 5

**Summary:**

The paper attempts to augment mechanisms for collaborative debate by getting debaters to state their reported uncertainty about a question. The paper explores three different methods for reporting uncertainties and then three different methods for aggregating this uncertainties into the judge (how to use the reported uncertainties).

**Strengths:**

The paper introduces an interesting novel mechanism for interpreting reported uncertainties from other models.
The paper is well written and structured well.

**Weaknesses:**

1) I’m not sure any of the results in Table 1 or Table 2 are statistically significant. On dataset sizes of 100 data points and without reported uncertainty, if I naively calculate the Standard Error of Mean (assuming results are binomial distributed (0 or 1)), then all results have overlapping confidence intervals. I’d advise running with more data or, if constrained, k-fold validation. Furthermore as generations are stochastic (at Temp=1) it would be good to run repeats anyway to clarify your reported results are good estimators of performance.


2) I’m unsure if the attention-masked method is really suitable. You either live in the land where the model is a black box (such as an API), or the model is a white box (and you can alter the weights and the attention mask). If you propose methods in the second approach, then surely, under minimal training, judges will pick up on the debater's uncertainties.


3) I think key components of the debate literature are missing:
Irving et al (AI safety via Debate)
Khan et al (Debating with more perusasive LLMs leads to more truthful outcomes)


4) I think the use of the oracle baseline is misleading here - in the situation where you have a perfect verifier in any of the debaters - we’d expect performance to be really high. This should be the upperbound for all methods, but is not an extrapolation of what models can achieve.


5) Some wording is interesting is concerning that a finding from the paper in section 5.2 is “The best-performing uncertainty metric was the Oracle metric.” - this is surely true by construction?

**Questions:**

1) How is the relevance score for TokenSAR generated?

2) I think using the Oracle should actually be used as an upper bound for performance of each aggregation (or uncertainty combination) method. I think this would change table 1 results and suggest aggregating with

3) In figure 4, the x-axis is the AUROC for the uncertainty metric - can you clarify which reported uncertainty this (is it the final judge). Also plotting uncertainty here (on the y axis would be useful). I also find it worrying that without oracle, all these trends are much weaker.

---

### Note · Authors · 2024-11-17

I have read and agree with the venue's withdrawal policy on behalf of myself and my co-authors.